# Evaluation of predictions of the stochastic model of organelle production based on exact distributions

C Jeremy Craven*

Department of Molecular Biology and Biotechnology, University of Sheffield, Sheffield, United Kingdom

**Abstract** We present a reanalysis of the stochastic model of organelle production and show that the equilibrium distributions for the organelle numbers predicted by this model can be readily calculated in three different scenarios. These three distributions can be identified as standard distributions, and the corresponding exact formulae for their mean and variance can therefore be used in further analysis. This removes the need to rely on stochastic simulations or approximate formulae (derived using the fluctuation dissipation theorem). These calculations allow for further analysis of the predictions of the model. On the basis of this we question the extent to which the model can be used to conclude that peroxisome biogenesis is dominated by de novo production when *Saccharomyces cerevisiae* cells are grown on glucose medium.

## Introduction

Recently a model was presented in which the variation of numbers of a particular type of organelle (Golgi apparatus, vacuoles or peroxisomes) observed in cells was proposed as a diagnostic indicator of the relative importance of different processes by which organelles can be formed and destroyed (*Mukherji and O'Shea, 2014*; see *Mukherji and O'Shea, 2015* for a correction). Here we re-examine the mathematical analysis of this model and show that further insight can be gained from considering exact calculations of the equilibrium distributions. For conciseness we will refer to the model, and the analysis in the associated paper (*Mukherji and O'Shea, 2014*), by the abbreviation SMOP (stochastic model of organelle production).

## Analysis

### The SMOP model in the context of "birth and death" models

In the SMOP model, four processes are envisaged for the production and destruction of organelles: de novo synthesis, fission, fusion and decay. These four processes are characterised by one rate constant each, defined in the SMOP paper as $k_{de\ novo}$, $k_{fission}$, $k_{fusion}$, and $\gamma$. Following the definitions in the SMOP paper, the probabilities of each of the four processes occurring in the next small time period $\delta t$ are given in *Table 1*. We also include in this table the total rate of each process that would be observed instantaneously in a large population of $N$ cells.

Models involving processes of this type are generically termed "birth and death" processes and have a very long history of analysis in the context both of the life sciences (e.g. evolution; *Yule, 1924*) and in the context of physical processes (e.g. detection of cosmic rays; *Furry, 1937*). Accessible discussions can be found in several books (*Bailey, 1990*, which is a reissue of the classic text from 1964; *Taylor and Karlin, 1998*). In such analyses, the three processes of de novo production, production by fission, and loss by first order decay are often termed immigration, birth and death,

*For correspondence: c.j.craven@sheffield.ac.uk

**Competing interests:** The author declares that no competing interests exist.

**eLife digest** Any cell that has a nucleus also contains a number of subcellular structures called organelles. The number of organelles inside a cell increases when new organelles are made from scratch (a process known as de novo synthesis), or when an existing organelle divides to produce two organelles in a process called fission. And the number of organelles decreases when an existing organelle decays, or when two organelles fuse together to become one organelle. The actual number of organelles of a particular type inside a cell results from a balance between these creative and destructive processes.

Last year researchers at Harvard University developed a model that treats the processes of organelle creation and destruction as if they were chemical reactions, and then used their model to make predictions about the budding yeast *S. cerevisiae* in three scenarios. The Harvard researchers had to use a number of approximations to make these predictions.

Now Jeremy Craven has derived exact solutions to the model for these three scenarios. The exact solutions call into question some aspects of the model, notably the prediction that the production of new peroxisomes – organelles that are involved in breaking down fatty acids and other compounds – is dominated by fission when the yeast cells are grown on a substance called oleate, and by de novo synthesis when they are grown on glucose. Craven's analysis also highlights the need for quantitative time-course imaging data to test theoretical models of dynamic processes in cells.

respectively. Immigration is used for a process that increases the number of individuals but does not require any other individuals already to be present. Birth is the process by which one individual gives rise to a second individual. Death is a process by which a particular individual is lost from a population with a probability that is independent of any other members of the population. Analyses including a fusion term are much less common.

As there are a considerable number of possible combinations of the four processes that might be active, we will use a notation here to define a model by listing in curly brackets the active production processes, followed by the active destruction processes, separated by a semi-colon. Any process that is not mentioned has a rate constant of zero. Thus the model with de novo, fission and decay terms would be denoted {de novo, fission; decay}.

During single cell simulations based upon the equations in *Table 1* the number of organelles will fluctuate (*Figure 1A*) and one can ask what fraction of time, $f_n$, does a cell spend having $n = 0$, $n = 1$, $n = 2$, etc. organelles. This is equivalent to asking what fraction of a large ensemble of cells have $n = 0$, $n = 1$, $n = 2$, etc. organelles at one moment in time. In treatments of stochastic systems, the values of $f_n$ would normally be described as the probability distribution for a cell having $n$ organelles. In terms of a population of cells it can be described as a population distribution.

**Table 1.** Definition of terms in the SMOP model

| Process | Probability of process occurring in next $\delta t$ in a particular cell containing $n$ organelles[1] | Process changes $n$ by | Rate of process in sample of $N$ cells[2] |
|---|---|---|---|
| De novo | $k_{de\ novo}\delta t$ | 1 | $k_{de\ novo}f_n N$ |
| Fission | $k_{fission}n\delta t$ | 1 | $k_{fission}nf_n N$ |
| Decay | $\gamma n\delta t$ | -1 | $\gamma nf_n N$ |
| Fusion | $k_{fusion}n(n-1)\delta t$ | -1 | $k_{fusion}n(n-1)f_n N$ |

[1]For example, if $k_{fission} = 0.02$ then the probability of a cell with $n = 2$ organelles undergoing a fission event in the next 0.1 time units = 0.02x2x0.1 = 0.004.

[2]$f_n$ is the fraction of cells having $n$ organelles. For example, if 23% of the cells have 2 organelles then $f_2 = 0.23$. If, in a population of 1000 cells, $f_2 = 0.23$ and $k_{fission} = 0.02$ then the rate of cells changing from having $n = 2$ to $n = 3$ organelles at any one moment due to fission would be 0.02x2x0.23x1000 = 9.2 cells per time unit.

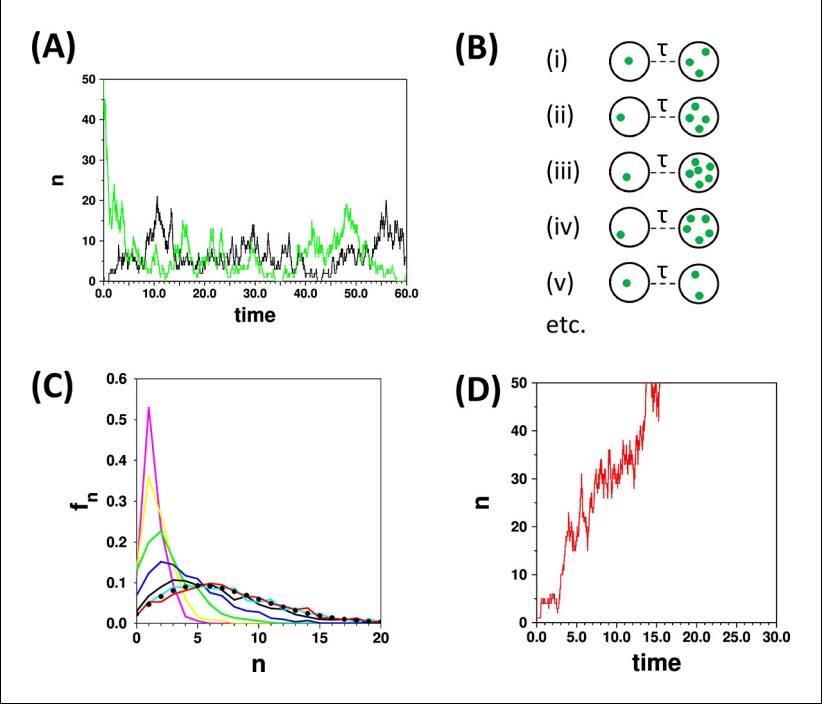

**Figure 1.** The concept of the limiting distribution in a stochastic system. (A) The traces show simulations run with the parameters {$k_{de\ novo}$ = 2.0, $k_{fission}$ = 0.9; $\gamma$ = 1.0, $k_{fusion}$ = 0.02}, starting from $n$ = 0 (black trace) and $n$ = 50 (green trace). Both simulations "settle down" to stochastic fluctuations about a mean value of <$n$> = 7.1 (B) Schematic representation of a set of cells that are all initialised to $n$ = 1 at time $t$ = 0, and are observed at a time $t$ = $\tau$. (C) Distributions calculated with the same parameters as in (A) for a set of 1000 cells as in (B), calculated for $\tau$ = 0.2 (magenta), 0.4 (yellow), 1.0 (green), 2.0 (blue), 5.0 (black), 10.0 (cyan), 15.0 (red). The curves for $\tau$ = 10.0 and $\tau$ = 15.0 become very similar as they approach the limiting distribution. These two curves match closely the result (filled black circles) of applying the recurrence relation (*Equation 2*; Appendix 2). (D) The red trace is a time course for parameters {$k_{de\ novo}$ = 2.0, $k_{fission}$ = 1.1; $\gamma$ = 1.0, $k_{fusion}$ = 0}. Since $k_{fission}$ > $\gamma$, and $k_{fusion}$ = 0, then $n$ diverges; in such a case there is no limiting distribution.

The simulations in *Figure 1A* illustrate the important point that a simulation is always started from some arbitrary starting point, and that a period of time must elapse before the simulations can be considered to be independent of this starting point. If the distribution $f_n$ is evaluated at different times after the starting point of simulations (*Figure 1C*) then different distributions are obtained; hence the distribution is "time dependent". As the probability (or population) distribution varies in time (and since the system can in principle be started from any state) then it is not strictly possible to talk of "the distribution" for a stochastic system. However in many birth and death models the system will settle down to a limiting distribution, independent of the starting states of the cell(s) as in the cyan and red curves in *Figure 1C*. Such a situation corresponds to a state of dynamic equilibrium that is familiar from chemical kinetics. Thus the terms limiting, equilibrium (or steady state) distributions can be used in this context interchangeably. The conditions for the models studied here to have limiting distributions is discussed further below; an example of a set of parameters for which a SMOP model does not yield a limiting distribution is shown in *Figure 1D*.

Assuming that a limiting probability distribution does exist there are two basic sampling methods by which it can be measured, irrespective of whether one is making experimental observations or performing simulations. One method is to note $n$ at a set of time points for a single cell (such as points drawn from the trajectories in *Figure 1A*), and the other is to take a large number of cells at one point in time, and measure $n$ across this ensemble of cells (*Figure 1B,C*). The former would require a time dependent set of observations that may be difficult to obtain experimentally. The latter approach is equivalent to making experimental observations on a large field of view of cells, or of making repeated simulations. However there is a large caveat that when the measurements are

made one must be convinced that the cells have had "long enough to reach equilibrium" since the last significant perturbation to the system. If this is not the case then the distribution measured will be contaminated with contributions from non-equilibrium distributions (such as from the magenta, yellow, green and blue curves in *Figure 1C*). Perturbations include the choice of an arbitrary starting point in simulations, and effects such as cell division and change of growth conditions in experimental data.

## Does the SMOP analysis imply equilibrium distributions?

It is implicitly assumed in the SMOP paper that the populations are to be considered to be at equilibrium (or that the probability distributions are in their limiting form) for all three cases of the analysis via simulations, from experimental data, or via the fluctuation dissipation theorem; a comment has been added to the original articles to clarify this assumption for the simulations (see the comment dated November 23, 2015 on *Mukherji and O'Shea, 2014*). For the experimental data this seems a reasonable assumption, although dynamic population data are really required to fully settle this issue. The fluctuation dissipation theorem method implicitly assumes a steady state (*Paulsson, 2005*).

## Derivation of recurrence relation for the distribution of organelles in the {de novo, fission; decay, fusion} model

By applying an equilibrium condition it is straightforward to derive precise relations for the distributions in the three scenarios considered in the original SMOP paper, and hence avoid the approximations introduced by the use of the fluctuation dissipation theorem.

At equilibrium, the rate at which the population gains cells with $n + 1$ organelles due to cells with $n$ organelles gaining one organelle must equal the rate at which the cells with $n + 1$ organelles lose one organelle. The reasoning is the same as for standard treatments of dynamic equilibrium between two states (as in a chemical reaction), and the complete justification of this when there are multiple states (i.e. cells with $n = 0$, $n = 1$, $n = 2$, etc., organelles) is given in Appendix 1.

Thus at equilibrium

$$(k_{de\ novo} + k_{fission}n)f_n N = (\gamma + k_{fusion}n)(n + 1)f_{n+1}N \tag{1}$$

which gives

$$f_{n+1} = \frac{(k_{de\ novo} + k_{fission}n)}{(\gamma + k_{fusion}n)(n + 1)}f_n \tag{2}$$

From *Equation 2* the *exact* distribution of organelle numbers at equilibrium can be calculated for a model involving *any* combination of the four processes, without recourse to random number based simulations and the attendant issues of ensuring adequate sampling precision.

An explicit numerical example of the use of *Equation 2* to generate a distribution is given in Appendix 2. Briefly, an arbitrary value for $f_0$ is chosen; $f_1$ is then calculated from $f_0$; $f_2$ is calculated from $f_1$; $f_3$ is calculated from $f_2$; etc. Finally the entire distribution is normalised, which removes any dependence on the initial choice for $f_0$. *Equation 2* is often termed a recurrence relation (or sometimes recursion relation or difference equation) as it allows successive terms in a distribution to be calculated from earlier terms.

## Application of recurrence method to Golgi and vacuole models

The recurrence relation readily allows the derivation of precise distributions for the case of the model applied to Golgi ({de novo; decay}, Appendix 3) and vacuoles ({fission; fusion}, Appendix 4). For the Golgi, a Poisson distribution is obtained as the limiting distribution in accord with the SMOP analysis. However for vacuoles a truncated Poisson distribution is obtained, and not the shifted Poisson distribution that is reported in the SMOP analysis. Although the difference between these distributions is quite subtle (Appendix 4), the variation of Fano factor with <n> is significantly different: the Fano factor for the truncated Poisson approaches 1 much more rapidly (*Figure 2*, green curve) than for the shifted Poisson (*Figure 2*, black curve).

In *Figure 2*, it can be seen that the experimental values quoted in the SMOP analysis are in excellent agreement with the *incorrect* prediction, whilst the agreement with the corrected theoretical

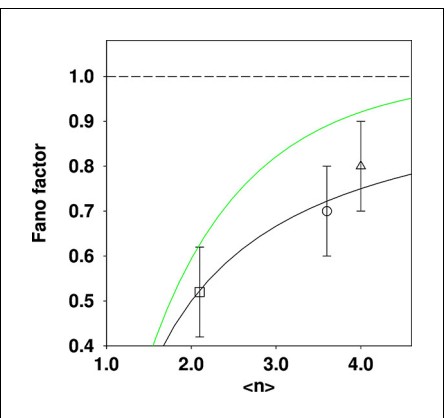

**Figure 2.** Comparison of reported Fano factors for vacuole populations, compared to two different theoretical expectations. Three data points quoted in the SMOP paper are plotted: □ Haploid (glucose); ○ Diploid (glucose); Δ Haploid (oleate). The solid black curve is the expectation from the shifted Poisson distribution (which is the incorrect distribution given the {fission; fusion} model) and the solid green curve is the expectation from the truncated Poisson distribution (which is the correct distribution given the model). The dashed line is for a Fano factor of 1. The solid curves were constructed by calculating a family of distributions and evaluating the mean and Fano factor.

prediction is much less good. This greatly weakens the argument that the SMOP model makes "quantitatively accurate predictions" or that it therefore correctly accounts for the behaviour of the vacuole population.

## Application of recurrence method to peroxisome models

Having established the value of analysing the SMOP model with the recurrence method, we move on to the case of peroxisomes.

### Recurrence relation demonstrates that {de novo, fission; decay} yields a negative binomial distribution

For peroxisomes, the primary model discussed in the SMOP analysis is a model in which peroxisomes can potentially form both de novo and by fission, and fusion is considered to be negligible. We show in Appendix 5 that such a {de novo, fission; decay} model has a limiting distribution that is a negative binomial distribution.

As a result, exact expressions (for all parameter values) are readily obtained (Appendix 5) for the mean and Fano factor for this model,

$$\langle n \rangle = \frac{k_{de\,novo}}{\gamma - k_{fission}} \tag{3}$$

$$\frac{\sigma^2}{\langle n \rangle} = \frac{\gamma}{\gamma - k_{fission}} \tag{4}$$

Combining *Equations 3,4* gives an alternative form for the Fano factor

$$\frac{\sigma^2}{\langle n \rangle} = 1 + \frac{k_{fission}\langle n \rangle}{k_{de\,novo}} \tag{5}$$

This latter equation is the form given in the SMOP analysis as the $k_{fusion} = 0$ limit of the approximate *Equation 1* of the SMOP paper. By explicitly applying the equilibrium assumption we have therefore shown that this expression is exact in this case for all values of the parameters.

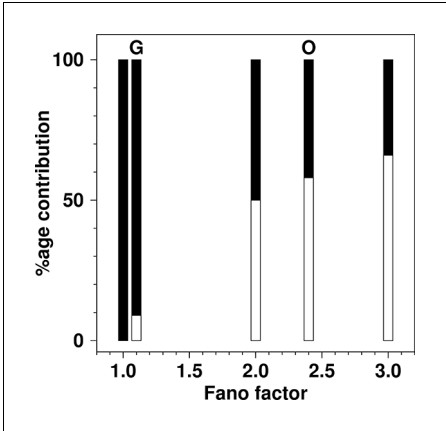

**Figure 3.** The percentage contribution to the total production rate from de novo production and from fission as a function of Fano factor in the {de novo, fission; decay} model. For each value of the Fano factor shown, a bar is drawn to represent the total production rate. The filled part of the bar represents the contribution from de novo production inferred from the model and the open part of the bar represents the inferred rate from fission. Bars are shown for Fano factors of: 1.0 (100% de novo); 1.1 (experimentally observed for glucose growth (**G**) in SMOP paper; 9% de novo, 91% fission); 2.0 (boundary value, where de novo and fission contributions are equal); 2.4 (experimentally reported value for oleate growth (**O**) in SMOP paper; 42% de novo, 58% fission); 3.0 (value required for fission rate to be double the de novo rate). The total production rate from de novo processes is simply $k_{de\ novo}$. The total rate from fission processes is $k_{fission}<n>$. The relative proportions of the two processes were calculated using **Equation 5**.

## Evaluation of Fano factor for the boundary marking equal de novo and fission rates

An attractive claim made for the SMOP analysis is that the value for the Fano factor can be used to distinguish between different modes of organelle production.

In **Figure 3D** of the original SMOP paper, a green dashed line was placed at $\sigma^2/\langle n \rangle = 1$ and stated as "marking the boundary between de novo synthesis and fission dominated organelle production". Thus, as presented in **Figure 3D** of the original version of the SMOP paper, the Fano factor for cells grown on oleate ($\sigma^2/\langle n \rangle = 2.4 \pm 0.2$) appears to lie significantly above the line where the rates are equal, and thus terms such as "fission dominated biogenesis" are used widely in the SMOP paper.

However, according to **Equation 5** the correct value with equal contributions from the two processes is $\sigma^2/\langle n \rangle = 2$. A correction to this effect has now been issued. The value of $\sigma^2/\langle n \rangle = 1$ corresponds to the extreme case of *zero* fission and production *solely* by de novo production (**Figure 3**). In the corrected version of **Figure 3D** in the SMOP paper it is clear that the inferred fission rate for growth on oleate is *only barely greater* than the de novo synthesis rate and the term "dominance" does not seem to be appropriate. It is clear from **Figure 3** that the inferred contribution from fission grows only rather slowly as the Fano factor increases above a value of two. In other words, for fission to be truly dominant a Fano factor would have to be observed that was far greater than any reported in the SMOP paper.

## Restrictions on the relationships between parameters in order for limiting distributions to exist

Since it is inherent in the SMOP analysis that the distributions are limiting/equilibrium distributions, it is important to consider whether a limiting distribution will ever be reached.

In the case of the peroxisome model, {de novo, fission; decay}, for the system to have a limiting or equilibrium distribution there is a strong restriction on parameters, namely that $k_{fission} < \gamma$. To see why this restriction exists, consider first the more simple {de novo; decay} model. Considering a single cell, the number of organelles will be limited by the fact that as $n$ grows larger then the decay

process (whose rate increases proportionally to $n$) will become increasingly more likely than the de novo formation process (which is independent of $n$). If the fission process is introduced then there is now a *production* term that also increases proportionally to $n$. If $k_{fission}$ exceeds $\gamma$ then the fission process will always exceed the decay process and the number of organelles will grow without limit.

The behaviour as $k_{fission}$ becomes similar to $\gamma$ can also be seen from *Equations 3,4* since if $k_{fission}$ is increased from zero until it becomes equal to $\gamma$ then $<n>$ and $\sigma^2/\langle n \rangle$ both diverge. Thus at first sight there appear to be two different ways in which the Fano factor can become very large. One way, as extensively discussed in the SMOP analysis and above, is if the mean rate $k_{fission}<n>$ becomes large compared to the mean rate $k_{de\ novo}$. The other way, that becomes clear from our analysis, is if the rate constant $k_{fission}$ becomes similar to the rate constant $\gamma$. The connection between these two relations is that $<n>$ is a function of all three rate constants. This emphasises the hidden complexity of the interplay of the parameters in this model.

## Discussion

Our motivation for this in depth analysis of the SMOP method was sparked by the claim that it could differentiate between fission and fusion dominated mechanisms of peroxisome biogenesis. That the distribution of numbers of organelles can give a clue to the mechanisms by which organelles are formed is a very elegant idea, and the SMOP analysis combines this idea with a very simple kinetic model. As we explored the system further we realised that the fluctuation dissipation theorem result was not necessary for analysis of the system, and that enforcing the equilibrium condition, that was implicit in the work already, greatly simplified the analysis.

There are a number of factors that cause us to question the utility of the SMOP model. A main piece of evidence for the correctness of the model was the agreement of the experimentally observed Fano factors for the vacuole data with those from the model. We have shown this agreement to be much less perfect than originally demonstrated. We have also shown that there is a strong interplay between different parameters in the model. This means that the agreement of experimental data with the model is not as compelling as originally presented and that the interpretation of experimental observations back to mechanistic conclusions is open to question. We hope that our analysis will stimulate discussion as to whether, for instance, the SMOP model captures the key features of the underlying processes and is just lacking some details; or whether the model fundamentally lacks key aspects of feedback. The assumption that observations of cells grown in batch culture faithfully report equilibrium distributions also requires further verification.

A key conclusion of the SMOP analysis is that the contribution of fission to peroxisome biogenesis is negligible (<10%) when yeast cells are grown on glucose, but "dominant" when they are grown on oleate. This is an area of some contention (*Hoepfner et al, 2005*; *Motley and Hettema, 2007*), and a recent model relied on fission of peroxisomes during organelle inheritance as the proliferation mechanism (*Knoblach et al, 2013*). Our analysis has shown that the term "dominant" is misleading, and that the data reported for haploid cells grown on oleate indicates approximately equal contributions from the two processes.

Nevertheless the model does suggest that the proportion of production by fission increases by about a factor of five on switching to oleate growth. Supporting evidence for this was provided by the observation of the reduction in the inferred fission contribution in cells grown on oleate in which the fission factors Vps1 or Dnm1 (or Fis1, an accessory factor of Dnm1) were deleted. On the other hand, no data were shown for glucose grown cells harbouring the same deletions. On glucose the Fano factor is reported in the SMOP analysis as 1.1, with $<n>$ = 3, and from *Equations 3,4* one obtains $k_{de\ novo}$ = 2.7$\gamma$ and $k_{fission}$ = 0.1$\gamma$. The model then implies that on deletion of the fission pathway (i.e. setting $k_{fission}$ = 0) then $<n>$ would only drop by 10%. Peroxisome count data has been reported recently (Fig S4, *Motley et al., 2015*), with values of $<n>$ = 4.9 in WT cells, $<n>$ = 1.5 in *vps1Δ* cells and $<n>$ = 1.2 in *vps1Δdnm1Δ* cells. The drops in peroxisome numbers in *vps1Δ* and *vps1Δdnm1Δ* cells are much greater than the 10% estimated above from the SMOP model. There is also peroxisome count data in *Kuravi et al. (2006)*, which gives $<n>$ = 1.6 (*WT*), $<n>$ = 1.2 (*vps1Δ*), $<n>$ = 1.7 (*dnm1Δ*), $<n>$ = 0.9 (*dnm1Δvps1Δ*). The drop in $<n>$ for the *dnm1Δvps1Δ* again conflicts with the idea that fission is such a small contributor to the biogenesis process. The discrepancies between these various data possibly arise from difficulties in quantifying peroxisome numbers, especially when cells contain a large number of small (and therefore low fluorescence) peroxisomes as

may be the case when fission is a strong contributor to biogenesis. The problem of peroxisome counts depending on the brightness of fluorescent markers has been commented on by *Jung et al.* (2010).

There is a continuing push for cell biology to become more quantitative, and to be subject to the use of rigorous models as are common in the physical sciences. Such a push raises significant challenges not only in terms of developing tractable models and justifying the underlying assumptions, but also in terms of the application of the model to complex experimental data. In particular this work highlights the need for greater accounting for detection limits and intensity distributions, as well of time dependent issues, in the reporting and analysis of organelle count data, if these are to be used to infer details of organelle biogenesis mechanisms.

## Materials and methods

All calculations were performed in python 2.7, running either via Cygwin under Windows 8.1, or Linux Mint 17.0. The code used for running stochastic simulations and for calculating distributions via *Equation 2* is given in *Source code 1,2*. Time units are arbitrary. The time step for the simulations in *Figure 1* was 0.0001 units. *Equation 2* can be reformulated in terms of three parameters, e.g. $k_{de\ novo}/\gamma$, $k_{fission}/\gamma$, $k_{fusion}/\gamma$; thus for example the parameters {$k_{de\ novo}$ = 2.0, $k_{fission}$ = 0.9; $\gamma$ = 1.0, $k_{fusion}$ = 0.02} and any uniformly scaled set of parameters (e.g. {$k_{de\ novo}$ = 4.0, $k_{fission}$ = 1.8; $\gamma$ = 2.0, $k_{fusion}$ = 0.04}) yield the same limiting distribution. As in the SMOP paper, the Fano factor is defined here as $\frac{\sigma^2}{\langle n \rangle}$.

## Acknowledgements

I thank Ewald Hettema, Paul Galvin and Ian Sudbery for helpful discussions, and Eleanor Stillman for pointing me towards helpful textbooks on stochastic processes.

## Additional information

### Funding

No external funding was received for this work.

### Author contributions

CJC, Conception and design, Acquisition of data, Analysis and interpretation of data, Drafting or revising the article, Wrote the Source code

## Additional files

### Supplementary files

• Source code 1. Code for calculating time courses via simulation.

• Source code 2. Code for calculating distributions via the recurrence relation.

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

## Appendix 1: Detailed balance

Consider an ensemble of cells, where cells with for example $n = 3$ organelles can only arise from cells with $n = 2$ or $n = 4$ organelles.

We can represent the kinetic processes thus

| | $R_{n=1 \to n=0}$ | | $R_{n=2 \to n=1}$ | | $R_{n=3 \to n=2}$ | | $R_{n=4 \to n=3}$ | | $R_{n=5 \to n=4}$ | |
|---|---|---|---|---|---|---|---|---|---|---|
| $n = 0$ | $\leftrightarrows$ | $n = 1$ | $\leftrightarrows$ | $n = 2$ | $\leftrightarrows$ | $n = 3$ | $\leftrightarrows$ | $n = 4$ | $\leftrightarrows$ | etc. |
| | $R_{n=0 \to n=1}$ | | $R_{n=1 \to n=2}$ | | $R_{n=2 \to n=3}$ | | $R_{n=3 \to n=4}$ | | $R_{n=4 \to n=5}$ | |

Where for example $R_{n=2 \to n=3}$ is the rate of formation of cells with $n = 3$ due to cells with $n = 2$ gaining one organelle, and $R_{n=3 \to n=2}$ is the rate of loss of cells with $n = 3$ due to such cells losing one organelle.

If the system is in equilibrium (*i.e.* the values of $f_0$, $f_1$, $f_2$, etc. are not changing) then a form of the principle of detailed balance applies. The principle of detailed balance asserts that for instance

$$R_{n=3 \to n=2} = R_{n=2 \to n=3} \tag{6}$$

To show this, consider first the state with $n = 0$. At equilibrium the rate of formation must equal the rate of loss,

$$R_{n=1 \to n=0} = R_{n=0 \to n=1} \tag{7}$$

Now considering the state with $n = 1$, the equivalent relation contains more terms

$$R_{n=0 \to n=1} + R_{n=2 \to n=1} = R_{n=1 \to n=0} + R_{n=1 \to n=2} \tag{8}$$

But, crucially, by substituting in from **Equation 7** this simplifies to

$$R_{n=2 \to n=1} = R_{n=1 \to n=2} \tag{9}$$

And by proceeding along the chain the general relationship emerges that

$$R_{n=i \to n=i-1} = R_{n=i-1 \to n=i} \tag{10}$$

For $i = 1, 2, 3$, etc.

In other words at equilibrium every individual reversible process must have balancing rates. The full "principle of detailed balance" for molecular systems contains some greater subtleties related to thermodynamics, that allow the principle to be applied to more complex networks.

## Appendix 2: Recurrence example

The recurrence relation can be applied in an algebraic form as in the analysis of the various models in the paper, but can also be used to rapidly calculate numerical distributions directly.

As an example, consider the model $\{k_{de\ novo} = 2.0, k_{fission} = 0.333; \gamma = 1.0, k_{fusion} = 0.0333\}$

*Equation 2* becomes

$$f_{n+1} = \frac{(2.0 + 0.333n)}{(1.0 + 0.0333n)(n+1)} f_n \tag{11}$$

We first calculate provisional (un-normalised) values for $f_0$, $f_1$, $f_2$, etc.

First set $f_0 = 1$. This is an arbitrary choice and the final results do not depend upon it.

Using *Equation 11* with $n = 0$,

$$f_1 = \frac{(2.0 + 0)}{(1.0 + 0)(1)} \times 1.0 = 2.0 \tag{12}$$

Then using *Equation 11* with $n = 1$, and the above value for $f_1$,

$$f_2 = \frac{(2.0 + 0.333)}{(1.0 + 0.0333)(2)} \times 2.0 = 2.26 \tag{13}$$

And again using *Equation 11* with $n = 2$, and the above value for $f_2$,

$$f_3 = \frac{(2.0 + 0.666)}{(1.0 + 0.0666)(3)} \times 2.26 = 1.88 \tag{14}$$

This procedure can be repeated to calculate any number of terms.

For a model with a limiting distribution then for large $n$ the $f_n$ will become very small, for instance here $f_{20} = 0.00000008$, and so it is not necessary to calculate more terms.

The calculated terms are then summed to give a normalising factor ($NF$),

$$NF = 1.0 + 2.0 + 2.26 + 1.88 + \cdots + 0.00000008 = 9.89 \tag{15}$$

The provisional values of $f_0$, $f_1$, $f_2$, etc. are divided by $NF$, which ensures that the fractional populations sum to one, giving

$f_0 = 0.10$, $f_1 = 0.20$, $f_2 = 0.23$, $f_3 = 0.19$, $f_4 = 0.13$, $f_5 = 0.08$, $f_6 = 0.04$, etc.

Ideally it should be checked with a longer sequence that, to sufficient precision, the results are independent of the number of terms calculated.

## Appendix 3: Application of recurrence relation to {de novo; decay} model

In the case of the Golgi, the model proposed in the SMOP analysis is a {de novo; decay} model. This model is also often termed an immigration death process. For this model *Equation 2* becomes

$$f_{n+1} = \frac{k_{de\ novo}}{\gamma(n+1)} f_n \tag{16}$$

So we have

$$f_1 = \frac{k_{de\ novo}}{\gamma} f_0 \tag{17}$$

$$f_2 = \frac{1}{2}\frac{k_{de\ novo}}{\gamma} f_1 = \frac{1}{2}\left(\frac{k_{de\ novo}}{\gamma}\right)^2 f_0 \tag{18}$$

$$f_3 = \frac{1}{3}\left(\frac{k_{de\ novo}}{\gamma}\right) f_2 = \frac{1}{6}\left(\frac{k_{de\ novo}}{\gamma}\right)^3 f_0 \tag{19}$$

etc., and so in general

$$f_n = \frac{1}{n!}\left(\frac{k_{de\ novo}}{\gamma}\right)^n f_0 \tag{20}$$

To ensure that the sum of all $f_n$ equates to one, it is simple to show from the power law definition of the exponential function that $f_0 = \exp(-k_{de\ novo}/\gamma)$, and hence

$$f_n = \exp\left(-\frac{k_{de\ novo}}{\gamma}\right)\frac{1}{n!}\left(\frac{k_{de\ novo}}{\gamma}\right)^n \tag{21}$$

Thus the limiting distribution is a Poisson distribution as stated without proof in the SMOP paper. The mean is given by $\langle n \rangle = k_{de\ novo}/\gamma$.

## Appendix 4: Application of recurrence relation to {fission; fusion} model

In the case of vacuoles, the model proposed in the SMOP analysis is a {fission, fusion} model. For this model *Equation 2* becomes

$$f_{n+1} = \frac{k_{fission}}{k_{fusion}(n+1)} f_n \tag{22}$$

*Equations 16,22* are identical except for the replacement of the ratio $k_{de\ novo}/\gamma$ by $k_{fission}/k_{fusion}$ and therefore *Equation 22* yields a form of Poisson distribution. It is not a standard Poisson distribution since, as noted in the SMOP paper, such a model cannot reach a state with $n = 0$ from a state with $n > 0$ since the fusion probability is zero when $n = 1$. This means that $f_0 = 0$. The rest of the distribution has the same form as for a regular Poisson distribution (*Equation 23*), except for a slightly different overall normalising factor. This distribution is known as a *truncated* (or sometimes *zero-truncated*) Poisson distribution (*Equation 24*). We have checked this analysis by direct calculation and simulation using the programs in *Source code 1,2*.

The relevant distributions can be defined as (*Appendix 4 Figure 1*)

Poisson:

$$f_n = \exp(-\lambda)\frac{1}{n!}(\lambda)^n, \; n = 0,1,2,3,\ldots \tag{23}$$

Truncated Poisson:

$$f_n = \frac{\exp(-\lambda)}{1-\exp(-\lambda)}\frac{1}{n!}(\lambda)^n, \; n = 1,2,3,\ldots$$
$$= 0 \qquad\qquad\qquad\qquad , \; n = 0 \tag{24}$$

Shifted Poisson:

$$f_n = \exp(-\lambda)\frac{1}{(n-1)!}(\lambda)^{n-1}, \; n = 1,2,3,\ldots$$
$$= 0 \qquad\qquad\qquad\qquad , \; n = 0 \tag{25}$$

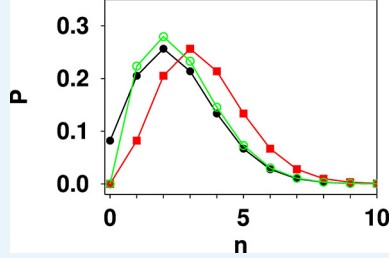

**Appendix 4 Figure 1.** Comparison of Poisson (black filled circles), shifted Poisson (red filled squares), and truncated Poisson (green open circles) distributions calculated for $\lambda = 2.5$. The values in a shifted Poisson distribution are the same as in the corresponding Poisson distribution, however the whole distribution is shifted to the right by one unit in *n*. The value at $n = 0$ is set to zero. In a truncated Poisson the value at $n = 0$ is set to zero, and the rest of the distribution has the same functional form as the corresponding Poisson distribution. Due

to the loss of the $n = 0$ point then normalisation causes the remaining values to be slightly higher in the truncated Poisson distribution than in the corresponding Poisson distribution.

## Appendix 5: Application of recurrence relation to {de novo, fission; decay} model

In the case of peroxisomes, the model proposed in the SMOP analysis is a {de novo, fission; decay} model. For this model *Equation 2* becomes

$$f_{n+1} = \frac{(k_{de\ novo} + k_{fission} n)}{(\gamma)(n+1)} f_n \tag{26}$$

This has the same form as the recurrence relation for the negative binomial distribution. The negative binomial distribution is often expressed as the distribution for the number of successes in a set of trials (probability of success in each trial = *p*) before a certain number, *r*, of failures occurs, however the distribution arises in a wide variety of situations (*Kotz et al, 1989*; *Taylor and Karlin, 1998*; *Scheaffer and Young, 2010*; *Grimmett and Stirzaker, 2001*) and in general there is no restriction that *r* be an integer. The recurrence relation for the negative binomial distribution is,

$$f_{n+1} = \frac{(pr + pn)}{(n+1)} f_n \tag{27}$$

and the negative binomial distribution has mean $\langle n \rangle = \frac{pr}{(1-p)}$ and variance $\sigma^2 = \frac{pr}{(1-p)^2}$.

Comparing *Equations 26,27* we can equate $p = k_{fission}/\gamma$ and $r = k_{de\ novo}/k_{fission}$

Hence

$$\langle n \rangle = \frac{k_{de\ novo}}{(\gamma - k_{fission})} \tag{28}$$

and

$$\frac{\sigma^2}{\langle n \rangle} = \frac{\gamma}{\gamma - k_{fission}} \tag{29}$$

Combining *Equations 28,29* gives an alternative form for the Fano factor,

$$\frac{\sigma^2}{\langle n \rangle} = 1 + \frac{(k_{fission} \langle n \rangle)}{k_{de\ novo}} \tag{30}$$

We also note that it is possible to derive an expression for the variance (and Fano factor) for the full {de novo, fission; decay, fusion} model as follows.

The total net rate of formation of organelles across the whole sample of cells, $R_{tot}$, is given by

$$R_{tot} = k_{de\ novo} N + \sum_i f_i N k_{fission} i - \sum_i f_i N \gamma i - \sum_i f_i N k_{fusion} i(i-1) \tag{31}$$

$$= k_{de\ novo} N + (k_{fission} - \gamma + k_{fusion}) N \langle n \rangle - k_{fusion} N \langle n^2 \rangle \tag{32}$$

At equilibrium $R_{tot}$ is zero, hence we have

$$\langle n^2 \rangle = \frac{k_{de\ novo} + (k_{fission} - \gamma + k_{fusion})\langle n \rangle}{k_{fusion}} \tag{33}$$

Making use of the relationship $\left\langle (n - \langle n \rangle)^2 \right\rangle = \langle n^2 \rangle - \langle n \rangle^2$,

we have

$$\sigma^2 = \frac{k_{de\ novo} + (k_{fission} - \gamma + k_{fusion})\langle n \rangle}{k_{fusion}} - \langle n \rangle^2 \tag{34}$$

And thus we have the *exact* result that

$$\frac{\sigma^2}{\langle n \rangle} = \frac{k_{de\ novo}}{k_{fusion}\langle n \rangle} + \frac{(k_{fission} - \gamma + k_{fusion})}{k_{fusion}} - \langle n \rangle \tag{35}$$

This expression can only be applied when $k_{fusion} \neq 0$, and it is not evident that it provides significant insight, however it is useful for cross checking other algebraic and computational results. We have been unable to find a useful closed expression for <n>.

