## [Decision Letter]

Thank you for submitting your work entitled "A reanalysis of the stochastic model of organelle production" for peer review at *eLif*e. Your submission has been favorably evaluated by Vivek Malhotra (Senior Editor) and two reviewers.

The reviewers have discussed the reviews with one another and the Senior Editor has drafted this decision to help you prepare a revised submission.

Specifically, the reviewers have raised a concern that your arguments are too aggressive and it would be best to tone down your statements. For example, you refer to "a number of errors" in the previous paper. The reviewers feel that this is incorrect or at least overstates the case. The reviewers agree that there was a major error in the O'Shea paper, namely the use of the shifted rather than the truncated Poisson distribution. Other than this, the mathematical results reported in the previous paper are correct as far as they go. Your manuscript shows that certain expressions in the Mukherji and O'Shea paper are exact, even though O'Shea and colleagues had reported them as approximate. This cannot be considered an error. The use of the word "dominant" is misleading, but is not a mathematical error. An abbreviated summary of the reviews follows.

A) Assessment of the manuscript:

First, I can find nothing mathematically incorrect about the present manuscript. The arguments are sound. Unlike the Mukherji and O'Shea paper, results are explicitly derived and all steps are detailed so readers are convinced of the correctness of these results. The paper is written in a rather pedagogical manner, the various results presented here are basically textbook material. For instance, deriving the detailed balance condition, explaining the stochastic simulation code, deriving the Poisson distribution, etc. would be standard graduate-level material typically not reported in a physics paper.

I do think a good point made in the present manuscript is the issue of identifiability, i.e. is it possible to identify an underlying microscopic model based on certain macroscopic measurements? It is clear from the new analysis that many combinations of parameter values can "mix together" to produce indistinguishable Fano factors, etc. However, on its own I don't think this analysis would have been considered a strong contribution. It is only as a critique of the Mukherji and O'Shea paper that this manuscript might be considered.

B) The contribution of the Mukherji and O'Shea paper:

The Mukherji and O'Shea paper made three major contributions:

1) It put forward the idea that heterogeneity in organelle number could be used to distinguish competing models of organelle biogenesis.

2) It made predictions from different models of biogenesis.

3) It reported experiments and compared them with predictions.

C) The critique of the Mukherji and O'Shea paper:

There are two major concerns raised about the Mukherji and O'Shea paper:

1) Were the simulations and the experiments truly done so the equilibrium assumption is correct?

2) The use of the shifted rather than the truncated Poisson distribution.

There are also two minor concerns:

3) What is the basis for various approximate Fano factors reported in the Mukherji and O'Shea paper?

4) The use of the term "dominance" in many places gives the reader the wrong impression.

Let's analyse each of these in turn.

1) During the review of the Mukherji and O'Shea paper, the issue of cell division as a confounding factor was explicitly raised. The authors' response was to provide more detailed simulations in which they showed that the Fano factors did not vary greatly when partitioning of organelles during division was taken into account, and was approximately as observed when the only mode of organelle reduction was a first-order decay term. This argument was convincing, assuming that equilibrium had truly been reached. I will add that the idea mentioned in the present manuscript (subsection “Does the SMOP analysis imply equilibrium distributions?”) that only if the time to equilibrium is short compared to the cell cycle time will the limiting distribution be valid, is a matter of scope. If the model explicitly includes cell division as one of the processes, that larger model will itself reach a limiting distribution. I agree that Mukherji and O'Shea provided no evidence that equilibrium had been reached, either in the simulations or the experiments, and were not requested to by the reviewers. This condition is typically assumed to be correct, and rarely are authors asked for evidence.

2) Is there any evidence that the simulations of the Mukherji and O'Shea paper had not reached equilibrium? I would assume that Mukherji and O'Shea would have done the correct validation steps, and there is some evidence of that in their Figure 1 for example. However, it is then difficult to understand how the error about the truncated vs. the shifted Poisson distribution could have arisen. I admit that this point could have been discovered by the reviewers if they had explicitly repeated some of the reported calculations in the Mukherji and O'Shea paper, and I for one overlooked it. It is unclear how the exact stochastic simulations could be in agreement with an incorrect shifted Poisson prediction. In the Mukherji and O'Shea paper no simulation matching the shifted Poisson prediction is given. Perhaps this simulation was never done, otherwise Mukherji and O'Shea would have discovered the discrepancy.

3) For certain expressions in the Mukherji and O'Shea paper, which arise out of an approximate calculation using the fluctuation-dissipation theorem (subsection “Incorrect analysis of cases with *k_fusion_* ≠ 0” of this manuscript), it is not explained how they were derived. This is a valid but minor concern; Mukherji and O'Shea would likely be able to provide the derivation. The point that the approximations were not necessary given the existence of an exact result is worth reporting but does not on its own detract from the original findings. In particular, many expressions derived in the Mukherji and O'Shea paper are correct, as shown in the present manuscript.

4) The misleading use of the word "dominant". There are scientific contexts where "dominance" is interchangeable with "greater than". E.g. in game theory, the word "dominance" is applied when the payoff from one strategy is just greater than from other strategy. Mukherji and O'Shea used the word "dominance" or "dominates" throughout their paper, but "contributed greater than" or some such term might have been more appropriate. I agree that this gives the reader the wrong impression, but does not, on its own, invalidate the results of Mukherji and O'Shea.

D) Is the critique valid?

I had said in my original review and I reiterate there: the ideas in the Mukherji and O'Shea paper were worth reporting, and I had hoped the original Mukherji and O'Shea paper would stimulate discussion and be improved upon with new analyses and measurements. The present manuscript in a sense validates this hope: discussion has indeed been stimulated. However, it is true that the original paper contained one major error not caught in the review process: use of the shifted rather than the truncated Poisson distribution. The present author argues that this weakens the central attraction of the previous paper, which was the strong agreement of theory and experiment. I would argue that the agreement itself, which was admittedly a striking feature, was not the central point on which the Mukherji and O'Shea paper stood. If the original paper had reported the correct distribution and highlighted the discrepancy with measured data, that would have also been food for thought.

Overall, I feel this is a more accessible and comprehensive theoretical analysis than the Mukherji and O'Shea paper provided, and serves to continue the original discussion.

---

## [Author Response]

*Specifically, the reviewers have raised a concern that your arguments are too aggressive and it would be best to tone down your statements.*

I have softened the language throughout the text. In addition, some of the references to parts of Figure 1 and Figure 4 have also been improved.

I have also taken the opportunity to add one sentence where I explain how I hope this will stimulate discussion of the underlying premises of the model (Discussion: “Our analysis has shown that the term […] contributions from the two processes”).

*For example, you refer to "a number of errors" in the previous paper. The reviewers feel that this is incorrect or at least overstates the case. The reviewers agree that there was a major error in the O'Shea paper, namely the use of the shifted rather than the truncated Poisson distribution. Other than this, the mathematical results reported in the previous paper are correct as far as they go.*

I have toned down some of these statements.

However the fact that the value of the Fano factor given by the expression 1 k_fission_<n>/k_de novo_ is equal to 2 when the terms k_fission_ <n> and k_de novo_ are equal and *not* to 1 as stated in the SMOP paper is a clear second mathematical error and was the reason why the k_fission_ term appeared so “dominant” in Figure 3, since the value for oleate growth is so close to 2 (almost within the stated error).

*Your manuscript shows that certain expressions in the Mukherji and O'Shea paper are exact, even though O'Shea and colleagues had reported them as approximate. This cannot be considered an error. The use of the word "dominant" is misleading, but is not a mathematical error. The detailed comments of the reviewers are pasted below.*

I have used the term “misleading” now so that I hope that will lead people to look more closely at the original arguments – and that that can lead to some of the healthy discussion that the reviewers refer to.

I am particularly keen to know the origin of the expression

σ2n = 2kde novo3kfissionn3kde novokfissionnsince I cannot find a way that it follows from their Equation 1 but maybe they have derived it via a different route. I am very happy to remove that comment if necessary, as the peroxisome model plus fusion does not seem to have much relevance given prevalent models of peroxisome biogenesis (detailed query is below).

A) Assessment of the manuscript:

First, I can find nothing mathematically incorrect about the present manuscript. The arguments are sound. Unlike the Mukherji and O'Shea paper, results are explicitly derived and all steps are detailed so readers are convinced of the correctness of these results. The paper is written in a rather pedagogical manner, the various results presented here are basically textbook material. For instance, deriving the detailed balance condition, explaining the stochastic simulation code, deriving the Poisson distribution, etc. would be standard graduate-level material typically not reported in a physics paper.

This area is so multi-disciplinary that I felt I should make the Discussion as accessible to as many workers in the field as possible. In addition, giving explicit examples help remove as much ambiguity as possible from definitions. I give the derivation of the Poisson distribution to show that it does arise from the recurrence relation: many people will be much more familiar with the Poisson distribution in the context of the number of successes of a large number of low-probability-of-success trials, which is really rather different from the birth-death type situation. I also give the Poisson (Golgi) case to show that my analysis does agree with the SMOP analysis in that case.

I give a full explanation of the principle of detailed balance as used here because the application here seems to me to be different from the thermodynamic principle of detailed balance. In a cellular system it seems there is no objection to a cycle having stable populations and yet sustaining a net flux round the cycle, whereas that is forbidden in closed thermodynamic systems. In the case of the present model it is the linear (and n>=0) nature of the set of linked reversible processes that lead to detailed balance. If there are accessible presentations of this material readily available and couched in sufficiently identical terms then I am more than happy to direct readers to those instead.

*I do think a good point made in the present manuscript is the issue of identifiability, i.e. is it possible to identify an underlying microscopic model based on certain macroscopic measurements? It is clear from the new analysis that many combinations of parameter values can "mix together" to produce indistinguishable Fano factors, etc. However, on its own I don't think this analysis would have been considered a strong contribution. It is only as a critique of the MO paper that this manuscript might be considered.*

In light of this comment, and the encouragement that this paper can form part of a healthy discussion, I have added a short paragraph discussing the implications of the underlying assumptions of the model (Discussion: “Our analysis has shown that the term […] contributions from the two processes”).

[…]

*C) The critique of the Mukherji and O'Shea paper:There are two major concerns raised about the Mukherji and O'Shea paper: 1) Were the simulations and the experiments truly done so the equilibrium assumption is correct? 2) The use of the shifted rather than the truncated Poisson distribution. There are also two minor concerns: 3) What is the basis for various approximate Fano factors reported in the Mukherji and O'Shea paper? 4) The use of the term "dominance" in many places gives the reader the wrong impression.*

I feel this list omits the crucial miscalculation of the value of the Fano factor for the case where the fission and de novo rates are equal in the peroxisome model, as I have discussed above.

*Let's analyse each of these in turn.*

*1) During the review of the Mukherji and O'Shea paper, the issue of cell division as a confounding factor was explicitly raised. The authors' response was to provide more detailed simulations in which they showed that the Fano factors did not vary greatly when partitioning of organelles during division was taken into account, and was approximately as observed when the only mode of organelle reduction was a first-order decay term. This argument was convincing, assuming that equilibrium had truly been reached. I will add that the idea mentioned in the present manuscript (subsection “Does the SMOP analysis imply equilibrium distributions?”) that only if the time to equilibrium is short compared to the cell cycle time will the limiting distribution be valid, is a matter of scope.If the model explicitly includes cell division as one of the processes, that larger model will itself reach a limiting distribution. I agree that Mukherji and O'Shea provided no evidence that equilibrium had been reached, either in the simulations or the experiments, and were not requested to by the reviewers. This condition is typically assumed to be correct, and rarely are authors asked for evidence.*

The cell division model is only included as a brief supplement to Figure 1 of the SMOP paper, and only for the Golgi ({de novo; fission}) model. Almost no details are given and that is significant because the issue of timescale becomes relevant in such a model. How does the “regular time interval” relate to the timescale of the various processes?

*2) Is there any evidence that the simulations of the Mukherji and O'Shea paper had not reached equilibrium? I would assume that Mukherji and O'Shea would have done the correct validation steps, and there is some evidence of that in their Figure 1 for example. However, it is then difficult to understand how the error about the truncated vs. the shifted Poisson distribution could have arisen. I admit that this point could have been discovered by the reviewers if they had explicitly repeated some of the reported calculations in the Mukherji and O'Shea paper, and I for one overlooked it. It is unclear how the exact stochastic simulations could be in agreement with an incorrect shifted Poisson prediction. In the Mukherji and O'Shea paper no simulation matching the shifted Poisson prediction is given. Perhaps this simulation was never done, otherwise Mukherji and O'Shea would have discovered the discrepancy.*

*3) For certain expressions in the Mukherji and O'Shea paper, which arise out of an approximate calculation using the fluctuation-dissipation theorem (subsection “Incorrect analysis of cases with k_fusion_ ≠ 0” of this manuscript), it is not explained how they were derived.This is a valid but minor concern; Mukherji and O'Shea would likely be able to provide the derivation.*

If it can be obtained I am very happy to drop this point. However I cannot see that it can follow from Equation 1. Equation 1 is

σ2n = 1kfusion(2n−1)γkfusion(n−1)γ−kfissionnkde novokfissionnIf the limit is looked for under the condition k_fusion_<(n(n-1))>) ~ γ <n>, it is not clear how to proceed because one immediately has terms in <n^2^>. (It appears that one can make some progress by then using variance=<n^2^>-<n>^2^ but that does not seem to lead to the quoted result).

If one instead writes the first fraction in the denominator of the left hand side of the above equation as X and try to solve simultaneously,

σ2n=1X−kfissionnkde novokfissionnalong with the quoted result

σ2n=2kde novo3kfissionn3kde novokfissionnthen one obtains an expression for X that depends on k_fission_<n> and k_de novo_ so again this route does not seem to lead to anywhere. However as <n> is inextricably linked to all parameters then maybe there is a hidden relationship that leads to the quoted result.

*The point that the approximations were not necessary given the existence of an exact result is worth reporting but does not on its own detract from the original findings. In particular, many expressions derived in the Mukherji and O'Shea paper are correct, as shown in the present manuscript.*

I agree, but if quantitative methods are to be adopted more within the mainstream then I think it behoves the modelers to make sure they keep things as simple as possible, otherwise it becomes the preserve of applied mathematician “wizards”. The recurrence method is much simpler than the FDT approach and should be accessible to more of the community.

*4) The misleading use of the word "dominant". There are scientific contexts where "dominance" is interchangeable with "greater than". E.g. in game theory, the word "dominance" is applied when the payoff from one strategy is just greater than from other strategy. Mukherji and O'Shea used the word "dominance" or "dominates" throughout their paper, but "contributed greater than" or some such term might have been more appropriate. I agree that this gives the reader the wrong impression, but does not, on its own, invalidate the results of Mukherji and O'Shea.*

I wrestled with this and sought the opinions of colleagues about what they would understand by “dominant”. If one rate were dominant over another then some people responded by saying it would imply a condition as extreme as differing by an order of magnitude. Personally I would want to see a factor of about three. Since the SMOP paper is a quantitative paper one would expect to see a statement of the ratio of the two rates, rather than the loose word dominant. In the present context the argument is made more complex by the use of a Fano factor of 1 rather than 2 as the boundary between equal rates. This is a significant difference when trying to evaluate the meaning of a Fano factor of 2.4 /-0.2. According to my analysis the rates are in the ratio 0.58:0.42, or roughly 1.4:1. Given the experimental and theoretical uncertainties this does not look to me like a secure case of dominance.

I agree that there are cases where dominant can mean “one just greater than the other”, and the game theory use is such a case. In politics an election vote distributed 0.58:0.42 would tend to be seen as indicating the dominance of a party.

Indeed if two rates act against each other then only a tiny “dominance” can lead to run away behaviour.

But here we are evaluating the relative magnitude of two rates that act in the “same direction” so I think that the requirement that dominance indicates a clear cut difference is valid.

I have amended my wording to “misleading” as suggested.

*D) Is the critique valid?*

*A4F1I had said in my original review and I reiterate there: the ideas in the Mukherji and O'Shea paper were worth reporting, and I had hoped the original Mukherji and O'Shea paper would stimulate discussion and be improved upon with new analyses and measurements. The present manuscript in a sense validates this hope: discussion has indeed been stimulated. However, it is true that the original paper contained one major error not caught in the review process: use of the shifted rather than the truncated Poisson distribution. The present author argues that this weakens the central attraction of the previous paper, which was the strong agreement of theory and experiment. I would argue that the agreement itself, which was admittedly a striking feature, was not the central point on which the Mukherji and O'Shea paper stood. If the original paper had reported the correct distribution and highlighted the discrepancy with measured data, that would have also been food for thought.*

The extremely close agreement of the observed and (incorrectly) predicted Fano factors seems to have provided a key link in the argument that the underlying model is a good one. That, in itself, is part of a fairly standard *inductive* scientific method with its attendant strengths and weaknesses. There was very little justification otherwise of the underlying model, and certain aspects of the model deserve more discussion.

For instance the use of a rate for fusion proportional to n(n-1) would seem appropriate if the organelles could be considered to be freely moving and well mixed as the molecules within a liquid. But whether it is appropriate for objects such as vacuoles seems harder to justify. It implies that the likelihood (within the next small delta-t) of a vacuole fusing with another when *n = 3* is *double* that when *n = 2*. One could envisage that when *n = 2* there could be a very similar degree of *total* contact area with other vacuoles as when *n = 3*, and so the fusion probability would be very similar. It would depend on vacuole size, composition and “packing”. I am not aware that sufficiently detailed analysis of such factors exists: it is certainly not quoted in the SMOP paper.

The observation of very good agreement with the model would reduce the impact of such an objection. However, in the absence of such agreement, then, such discussion is vital: is the model basically a good one but just lacking a little detail or is the model fundamentally lacking key aspects of feedback? I hope this will stimulate more discussion on these issues.